# The Weight Problem: Overview of the Most Common Concepts for Body Mass and Fat Distribution and Critical Consideration of Their Usefulness for Risk Assessment and Practice

**DOI:** 10.3390/ijerph182111070

**Published:** 2021-10-21

**Authors:** Dorothea Kesztyüs, Josefine Lampl, Tibor Kesztyüs

**Affiliations:** 1Department of Medical Informatics at the University Medical Centre Göttingen, Georg August University, Von-Siebold-Str. 3, 37075 Göttingen, Germany; tibor.kesztyues@med.uni-goettingen.de; 2General Practitioner Centre Arnold & Liffers, Albstr. 6, 89081 Jungingen, Germany; josi.lampl@googlemail.com

**Keywords:** anthropometry, weight indices, applicability, overweight and obesity, pandemic, health risk

## Abstract

The prevalence of obesity already reached epidemic proportions many years ago and more people may die from this pandemic than from COVID-19. However, the figures depend on which measure of fat mass is used. The determination of the associated health risk also depends on the applied measure. Therefore, we will examine the most common measures for their significance, their contribution to risk assessment and their applicability. The following categories are reported: indices of increased accumulation of body fat; weight indices and mortality; weight indices and risk of disease; normal weight obesity and normal weight abdominal obesity; metabolically healthy obesity; the obesity paradox. It appears that BMI is still the most common measure for determining weight categories, followed by measures of abdominal fat distribution. Newer measures, unlike BMI, take fat distribution into account but often lack validated cut-off values or have limited applicability. Given the high prevalence of obesity and the associated risk of disease and mortality, it is important for a targeted approach to identify risk groups and determine individual risk. Therefore, in addition to BMI, a measure of fat distribution should always be used to ensure that less obvious but risky manifestations such as normal weight obesity are identified.

## 1. Introduction

The COVID-19 pandemic has temporarily brought some increased media attention to the health risk posed by obesity. A systematic review identified three retrospective cohort studies that have independently reported a higher risk of severe disease progression in obese patients [1]. In the United States of America, where obesity prevalence is now approximately 42% of the adult population [2], more people at younger ages with obesity were severely affected by COVID-19 [3]. The underlying mechanisms are thought to be related to the multiple effects of primarily ectopic fat deposition on cardiovascular, metabolic, respiratory and immunologic functions [4]. These effects are also evident in the association of obesity with almost all noncommunicable diseases, which in 2019 represented 41% of deaths in low-income countries and up to 88% of deaths in high-income countries—91% in Central Europe; 88% in North America; and 74% worldwide [5]. Moreover, the rising incidence of overweight and obesity is also reflected in the prevalence of these noncommunicable diseases—coined a “slow-motion disaster” by the World Health Organization (WHO, Geneva, Switzerland) [6]. The increased risk of disease among overweight people therefore prompted the German Alliance against Noncommunicable Diseases (DANK) to recommend preventive measures, especially in the direction of weight control, to stop the “tsunami of chronic diseases” [7].

Obesity is by definition “an increase in body fat above normal” [8]. Classification is usually performed by applying the body mass index (BMI) following the categories specified by the WHO in 2001. Accordingly, overweight is defined as a BMI of 25 or higher, and obesity as a BMI of 30 or higher [8]. The vast majority of studies on associations between obesity and noncommunicable diseases are based on BMI criteria. The striking heterogeneity of obesity by BMI criteria in terms of cardiovascular and metabolic risks in many studies can probably be explained by fat distribution [9]. Therefore, the BMI concept is subject to increasing criticism because it neither takes fat distribution into account nor differentiates between fat and muscle mass. To date, there is no “gold standard” for measuring obesity, and BMI, the predominantly used measure, often fails to achieve correct classification [10]. On the basis of BMI, reliable estimates of obesity-attributable mortality are therefore difficult to obtain. Due to additional methodological issues, population-attributable fractions for obesity in terms of morbidity and mortality may be best considered as indicative of an association [11].

Based on this, some alternative concepts and measures have been added in recent decades to compensate for this deficiency. We will critically review BMI and the most important concepts of alternative determinations of excessive body mass or body fat mass and discuss their usefulness with regard to their informative value in terms of health risk. Emphasis is placed on the applicability of measures in clinical practice for the identification of high-risk patients—which needs to be possible at the local level, performed by general practitioners and family physicians, rather than in highly specialized clinical settings. Determining an individual patient′s risk of morbidity and mortality as reliably as possible by simple means is, after prevention, one of the most important means of managing the obesity pandemic.

## 2. Materials and Methods

In order to achieve a comprehensive overview and to validate the statements of this article not only by personal expert knowledge but also on a scientific basis, a thorough literature review was conducted. The databases Embase including Medline and the Web of Science were searched for articles on the definition and measurement of overweight, obesity and fat distribution, and their significance with regard to health risks and mortality. The main keywords used were “overweight”, “obesity”, “abdominal obesity”, “body fat”, “intra-abdominal fat”, “morbidity”, “mortality risk”, “anthropometry”, “diagnosis” and “epidemiology”. There was no time constraint applied to our literature search. The focus, already included in the search strategy, was specifically on comparative considerations of different indices with BMI as the standard measure, comparing as many indices as possible. For the main topics “mortality” and “disease risk”, only large, longitudinal population-based surveys, cohort studies, reviews and umbrella reviews with at least 5000 participants were included. Both sexes needed to be represented and mortality risk needed to reflect all-cause mortality or mortality associated with noncommunicable diseases, such as cardiovascular disease or cancer. With regard to the risk of disease, the focus was also on the most important noncommunicable diseases for which obesity represents a major risk; in addition to cancer, these are primarily cardio-metabolic diseases. Ethnicity plays an important role in anthropometry and the risk of disease or mortality [9,12], so we focused on Western populations, with only a few exceptions where information for these was lacking, and otherwise we would exceed the limits of this article. 

The references of the identified and included articles were searched for further information. With regard to the different ways of measuring fat percentage and distribution, emphasis was placed on their applicability and usefulness in identifying risks. Therefore, special methods for determining body fat percentage, such as measuring skinfold thickness using caliper forceps or hydrodensitometry, were not included because they require specially trained examiners or are too technically complex. Likewise, indices based on complex calculations were not presented.

## 3. Indices of Increased Accumulation of Body Fat

Searches in medical databases resulted in a plethora of articles with diverse methods for identifying unhealthy increases in or the ectopic deposition of body fat. Table 1 provides an overview of the most common approaches for which further information on risk, morbidity or mortality is also available.

BMI is a rather poor measure of increased body fat percentage as it does not differentiate between fat and muscle mass, which may lead to an underestimation of the prevalence of overweight and obesity [22]. A meta-analysis revealed that especially with regard to children, the diagnostic performance of the BMI is so limited that more than a quarter of all those with an elevated body fat percentage were not identified [23]. A systematic review with meta-analysis including a sample of 31,968 adult persons concluded that BMI cut-offs fail to detect half of individuals with excess %BF. The sensitivity therefore was 50%, while the specificity was 90% for detecting obesity [24]. Based on BMI data from 40,420 participants of the United States National Health and Nutrition Examination Survey (NHANES) 2005–2012, an estimated 75 million adults were misclassified in terms of cardio-metabolic health. The authors urge consideration of the unintended consequences of solely relying on BMI and improvement in the diagnostic tools for weight and cardio-metabolic health [25]. Therefore, the use of a complementary measure that takes fat distribution into account is often required. Apart from determining the percentage of body fat, these measures should explicitly include (intra-) abdominal fat since more metabolic activity occurs through the increased secretion of primarily endocrine factors in visceral fat compared to subcutaneous fat [26]. 

In their rationale for the selection of cut-off points for WC and WHR, the WHO considered different applications of sensitivity and specificity, e.g., the equivalence of sensitivity and specificity. Based on this, studies from different countries reported different cut-off points for several diseases, health risks or mortality, and it is a difficult task for the WHO to sum up a value for all [13]. The results of a meta-analysis of 32 studies showed a sex-specific sensitivity of the BMI of 51% and 50% and a specificity of 95% and 97% in females and males, respectively, for the detection of obesity. For waist circumference, the sensitivity was 62% and 57% and the specificity was 88% and 95% for females and males, respectively. Obesity was defined as a BMI ≥ 30 or a waist circumference ≥ 88 cm for females and ≥102 cm for males. The determination of the sensitivity and specificity was carried out in comparison with body fat measurements performed by computed tomography (CT), magnetic resonance tomography (MRT), dual X-ray absorptiometry (DXA) or ultrasound measurements. For the definition of obesity, the authors of this meta-analysis applied a body fat percentage of >35% in females and >25% in males because no generally validated cut-offs were available and the included studies used different values. Unfortunately, a calculation of sensitivity and specificity for the combined use of BMI and abdominal circumference was not available [27]. In a systematic review, the area under the curve of the receiver operator characteristic (AUROC) was 0.704, 0.693 and 0.671 for WHtR, WC and BMI, respectively, for the prediction of cardio-metabolic outcomes. A weighted mean cut-off of 0.5 for WhtR was valid for different populations and both sexes [14]. 

CT is a reference standard for measuring VAT, while alternative methods such as ultrasound and BIA show differences in sex and BMI categories [16]. A common agreement on cut-off points for VAT does not currently existing. A study comparing the correlation of VAT with WC and WHR concludes that WC and VAT agree regardless of sex or degree of obesity [15]. There are also no validated cut-offs for VFT to date. The cut-off values for SAD reported in Table 1 showed a sensitivity of 85% and 83% for females and males, respectively, and a corresponding specificity of 77% and 82% for predicting a 100 cm^2^ level of VAF derived from CT [18]. However, the question of a suitable cut-off for VAF still remains unresolved, and SAD values must therefore be viewed with caution. 

Based on ROC analyses, sex-specific %BF cut-off values for predicting obesity-related cardiovascular risk factors were determined with BIA in a study with 4735 participants, limited to an age range of 45–64 years [19]. The cut-off values of %BF depend on risk factors, age, sex and ethnicity, so the practical applicability of this concept for the definition of overweight and obesity is questionable. BIA also requires the use of different model parameters according to age, sex and level of physical activity, etc. to be reliable. Although there is some consensus that tomographic methods are the gold standard, they show differences between them; therefore, one completely accurate method has yet to be determined [28].

Finally, the BAI and ABSI are two newer indices that require some computational effort. The first claims to reflect %BF and was developed in a population study and showed a correlation of R = 0.85 between DXA-measured %BF and BAI in a validation study. The authors emphasized that no weighing scale was needed and the index was independent of sex and ethnicity. BAI was developed with data from 1277 Mexican Americans and validated with 223 African Americans [20]. The second new index combines BMI with WC and height as measures of body mass plus shape, adjusting the WC for weight and height, respectively. The ABSI is based on data from 12,105 adult participants from the NHANES 1999–2004 and predicts mortality risk better than BMI or WC alone. The ABSI correlation with mortality risk held for all age, sex and BMI categories as well as for White and Black—but not for Mexican ethnicities. The authors report quintiles of ABSI z-scores according to the relative mortality risk and recommend ABSI in addition to a low or high BMI in risk assessment [21]. 

With the exception of visceral fat (VAT, VFA, VFR) and %BF measurements, all indices presented here are easy to assess using simple tools such as a calibrated flat scale, stadiometer, metal measuring tape and abdominal caliper. For the more sophisticated calculation of BAI and ABSI, there are online calculation tools available through the Internet, but nothing is known about their reliability.

To date, for some of the indices for there were no validated cut-offs, we found that missing clear differentiation into areas of lower and increased risk strongly limits their usefulness for daily practice. In addition, some indices lack validation for sex, age or ethnicity.

## 4. Weight Indices and Mortality

Many epidemiological studies deal with different weight indices and their association with mortality. Table 2 contains basic information from large population-based studies.

A retrospective cohort study with approximately 247,000 multiracial participants showed that, essentially, an increased BMI also appears to pose an increased mortality risk, with a J-shaped relationship between BMI and mortality in which those with a BMI in the normal range show the lowest mortality [29]. Studies of body weight and mortality risk have to take the largest confounding factors, smoking and reverse causality into account, otherwise strong biases occur and the mortality risks of overweight and obesity are underestimated [43].

Some studies report differences in males and females in the risk prediction of indices of fat distribution which may also be associated with sex-related fat distribution patterns and the particular relevance of ectopic fat deposits, especially that of visceral fat [27,31,32,33,34,38,40,41,42]. Hence, sex should always be considered a confounding factor.

Apparently, the mortality risk of obesity attenuates with age, as a specific study of age-related differences revealed a mortality risk only for adults younger than 65 years, independent of measures of obesity [42]. This decline in mortality risk with age was also reported by another population-based study with over 2 million participants [44]. One possible explanation for this seemingly diminishing risk is a gradual decline in weight which begins to accelerate a few years before death, with a longer period before CVD and a shorter period before cancer death. Hence, weight loss reflects the consequences of disease and is therefore more apparent in older age—as is the prevalence of disease [45]. Therefore, the meaningful assessment of mortality risk due to excess body weight not only in older age can basically only be made against the background of existing diseases. 

The examination of 5540 NHANES participants who had never smoked, aged 50–84 years and whose maximum weight ever reached was recorded, revealed discrepancies in all-cause mortality between maximum BMI and BMI at the time of survey. Mortality attributable to maximum BMI was 33% and 5% at the time of survey, indicating a much stronger association with maximum BMI than with BMI at the time of the survey. This result could possibly indicate that the burden of obesity (in the US) is underestimated. Continuing, the author takes up the argument that the weak or inverse association between excess BMI and mortality in older adults found in some studies is due to disease-related weight loss [30]. For this reason, a person’s maximum weight to date should always be asked for when their current weight is recorded in order to better estimate the risk.

However, compared to BMI, measures that take into account fat distribution, especially abdominal fat, have greater predictive power. The results in this regard are heterogeneous in terms of which measure is most suitable, but overall indicators of abdominal obesity appear to be better predictors of mortality than BMI. The combined use of BMI and a measure of abdominal obesity is probably reasonable [32,35]. Although one study failed to find a significant difference between measures of overall fat distribution (BMI, %BF) and those of abdominal obesity (WC, WHR, WHtR) [41], the majority of studies support the recommendation that BMI measurement should be generally supplemented by a measure of fat distribution or abdominal obesity for a better risk assessment.

## 5. Weight Indices and Risk of Disease

Surprisingly, the number of studies on the risk of disease is significantly lower than the number of studies on mortality. This may be due to the fact that there are many cross-sectional studies that can only show associations but no risk of disease. Table 3 shows various prospective studies and an umbrella review on various diseases and the significance of overweight measures in this respect. 

In a prospective cohort study with 26,607 participants from Alberta, Canada, the risk of cancer was 33% increased for males and 22% for females with obesity based on BMI measures. However, the effect of BMI on all-cancer risk was substantially attenuated after the inclusion of WC as a continuous variable [46]. A comprehensive umbrella review of 204 meta-analyses of 371 cohort, 134 case-control and 2 cross-sectional studies found strong evidence of risk from adiposity for 11 cancers (oesophageal adenocarcinoma, multiple myeloma, cancers of the gastric cardia, colon, rectum, biliary tract system, pancreas, breast, endometrium, ovary, kidney). Although the underlying meta-analyses as a whole addressed seven indices (BMI, WC, hip circumference, WHR, weight, weight gain and weight loss), little information was provided on the cancer risk of abdominal obesity, probably because by far the largest part of the analyses was performed with respect to BMI, while WC and WHR were rarely considered [47]. In contrast, a population-based cohort study from Sweden comprising data from 27,557 participants compared several obesity indices to identify which showed the highest association with developing haematologic malignancies (HM). WC and ABSI were the best predictors for HM; all others, including BMI, showed no significant association. WC and WHR were also associated with higher risk for multiple myeloma, but none of the measures showed associations with myeloid malignancies or non-Hodgkin lymphoma [48]. The SAD, as a measure of intra-abdominal fat, was examined for its potential role as a predictor of liver disease in a Finish population-based study with 6636 participants. WHR performed best and WC as well as WhtR showed stronger associations than SAD, while BMI was non-significant [49]. The risk of stroke and its association with visceral obesity defined by WC, WHR and WhtR was examined in a multinational multi-cohort study with 54,717 participants. All measures, particularly WhtR, showed stronger associations than BMI, and were also strongly associated with a higher risk in those in the normal-weight BMI category [50]. Finally, the risk of first-onset cardiovascular disease (CVD) was studied separately and combined for BMI, WC and WHR with data from 221,934 individuals in 58 cohorts from 17 countries. The risk adjusted for age, sex and smoking status was similar for all three measures with a BMI slightly lower than WC and WHR. The medium and upper tertiles of WC and WHR showed a higher risk of incident CVD in the lowest tertile of BMI (<24.5). Further adjustment for additional intermediate risk factors (systolic blood pressure, diabetes, total and HDL cholesterol), for which information was available in 70% of participants, attenuated the association [51]. 

It is undisputed that all indices of increased body (fat) mass are associated with the diseases mentioned, and moreover, can act as predictors to varying degrees. Again, indicators of abdominal obesity often have a stronger predictive power and should be used in addition to BMI. For CVD risk, other well-known risk factors which in combination cause the metabolic syndrome can be used to predict risk.

## 6. Normal Weight Obesity and Normal Weight Abdominal Obesity

An often-overlooked risk group is that of people of normal weight with either an increased body fat percentage or abdominal obesity. People of normal weight by BMI definition may have metabolic characteristics and pathological disorders of obesity, the reasons for which are suspected to be mainly due to ectopic fat distribution, enlarged adipocytes and inflammation of adipose tissue [52,53]. An examination of 38,006 male participants, aged 40–75 years, from the prospective Health Professionals Follow-Up Study, revealed that in the normal BMI range, an unfavourable ratio of fat mass to lean mass is possibly associated with an increased risk of mortality [54]. An analysis of data from 6171 participants >20 years in the US Health and Nutrition Survey (NHANES) showed a high prevalence of cardio-metabolic dysregulation for a normal BMI in the highest tertile of %BF as compared to the lowest. After adjustment, a 2.2-fold increased risk for cardiovascular mortality for females in the highest tertile of %BF was detected [55]. The aim of an Australian prospective cohort study with 41,439 participants aged 27–76 was to determine the mortality risk and quantify deaths attributable to combinations of BMI and waist circumference, with abdominal obesity defined by the cut-off values of 88 cm for females and 102 cm for males. This almost consistently showed that normal weight abdominal obesity and the combination of obesity with abdominal obesity carried higher mortality risks compared to normal weight and obesity without increased waist circumference. Overall, the estimated proportion of all-cause and cardiovascular mortality attributable to waist circumference alone or the combination of waist circumference and BMI was higher compared with obesity defined by BMI alone [35]. 

Normal weight with abdominal obesity also presented with sex-independent increased risk factors for cardiovascular disease, such as hypertension, dyslipidaemia and type 2 diabetes compared to normal weight without abdominal obesity in a cross-sectional study of 117,163 Japanese adults aged 40–64 years [56]. Some studies provided evidence that people with normal BMI but abdominal obesity have an increased risk of both all-cause and cardiovascular mortality [57,58]. The increased mortality affects people with normal weight abdominal obesity defined by 18.5 ≤ BMI < 25 and WhtR ≥ 0.5 or WHR ≥ 0.85 for females and ≥0.90 for males; BMI alone does not identify this risk group in this analysis of biobank data. The prevalence of normal weight abdominal obesity was approximately 14% in the examined cohort of 6530 Australians aged 18–65 years [57]. Examining 15,184 adults aged 18–90 years from NHANES III showed that the increased mortality risk in those with normal weight abdominal obesity applies not only in comparison to normal weight without abdominal obesity, but also to BMI-defined overweight and obesity in males as well as in females [58].

There is likewise an association between fat distribution and the presence of cardiovascular risk factors which are also more prevalent in people with normal weight abdominal obesity [56,59]. Finally, while BMI and weight decline in older age, WC and visceral fat continue to increase [42]. This may lead to a higher proportion of normal weight abdominal obesity in the elderly, which should be considered in a risk assessment.

## 7. Metabolically Healthy Obesity

The risk of obesity-related comorbidities cannot be explained by the extent of obesity or fat distribution alone [53]. The phenomenon of obesity without cardio-metabolic impairment occurs more frequently in females and decreases with age [60]. A systematic review of 40 population-based studies identified a proportion of 35% metabolically healthy individuals among participants with obesity, albeit with considerable variance [61]. To date, there has been no uniform definition of the so-called “metabolically healthy obesity” apart from an existing basic consensus on the presence of a BMI ≥ 30 [53]. Often, the absence of cardio-metabolic disorders in the presence of obesity is used as a definition, and current proposals for criteria for a uniform basis include cut-off values for blood pressure, blood glucose, triglycerides, HDL and the exclusion of cardiovascular disease [53,62]. However, the risk of type 2 diabetes and cardiovascular disease is still higher in obesity without metabolic disorders compared with healthy normal weight [53]. The advantages of being metabolically healthy rather than unhealthily obese can be seen not only in lower insulin resistance but also in altered concentrations of other hormones and cytokines such as leptin, adiponectin and ghrelin [63]. Further differences relate to adipose tissue and the storage location, as in metabolically healthy obesity, the hyperplasia of small insulin-sensitive adipocytes predominantly occurs in subcutaneous fat depots. In metabolically unhealthy obesity, fat is increasingly ectopically stored in endocrine-active fat depots, for example, viscerally in the liver, or epicardially [64]. Here, the hypertrophy of adipocytes with deficient vascularisation may occur, leading to hypoxia, apoptosis, stress and inflammatory reactions in the affected tissue [53]. Most researchers agree that metabolically healthy obesity is a transient condition that sooner or later progresses into metabolically unhealthy obesity [53,62,63]. Therefore, the absence of cardio-metabolic dysfunction in obesity should not lead to delayed or omitted therapy. 

Overall, metabolically healthy obesity that is nonetheless associated with increased risk should be closely monitored as it develops.

## 8. The Obesity Paradox

A J-shaped relationship between BMI and mortality is often reported with a nadir in the BMI range of 21–25 [29,44]. In many studies, however, it is noticeable that patients with a higher BMI have better chances of survival. For example, although people with obesity are at a higher risk of heart failure, overweight and mild obesity appear to have a protective effect on disease-related mortality. The authors of this review see the inability of the BMI to differentiate between body fat and lean mass as a possible reason for the paradoxical correlations [65]. For coronary heart disease (CHD), there is also evidence from many studies of lower mortality in people with overweight and obesity compared to those with normal weight [66]. Here, the authors suspect the cause in the distribution of fat and the proportion of lean mass in different weight groups. They assume a high risk in a low body weight with a high percentage of fat as well as in abdominal obesity in general, whereas a high percentage of fat combined with a high lean mass or a gynecoid fat distribution carries a lower risk [66]. The evaluation of epidemiological data from 38,000 males with regard to body composition and mortality goes in a similar direction; here, the association between BMI and mortality was based on the ratio of lean body mass to fat mass [54]. A study on cancer with 175 participants showed the presence of the obesity paradox only when using the BMI, while sarcopenic obesity was associated with the worst prognosis [67]. Another investigation with data from over 500,000 people analysed mortality in relation to type 2 diabetes, CHD and cancer, based on different obesity indices (BMI, WC, %BF, WHR) [68]. In terms of BMI, the obesity paradox was observed in people with type 2 diabetes but not in people with CHD. It was pronounced in current smokers, absent in non-smokers and more pronounced in males than in females. Other measures of obesity provided less evidence of a paradox, but smoking status consistently influenced the relationship between obesity and mortality [68]. Instead of using the BMI at a fixed point in time, another approach used the maximum BMI in the observed period. Based on this, the percentage of mortality that can be attributed to overweight and obesity in 50–84-year-old non-smokers was 33%, compared to 5% for the conventional approach [30]. 

In the presence of coronary artery disease, it was clearly shown that measures of abdominal obesity are associated with mortality risk, whereas BMI alone mediates an apparently protective effect [69]. The obesity paradox appears to be predominantly a BMI paradox [66], and the validity of prospective studies of the association between mortality and BMI is limited by age- and disease-related weight loss [30].

The obesity paradox is not yet fully understood but there is some reasonable evidence that it is related to a lack of differentiation between fat and lean mass and is often influenced by smoking as a strong confounder. 

## 9. Conclusions

The risk of disease and mortality from increased body mass or body fat mass has been proven many times and is indisputable. In the review of large, population-based studies of the performance of various anthropometric indices, it became apparent that BMI used alone underestimates risk and miscategorises many individuals, particularly those with elevated risk. Therefore, a measure of fat distribution should always be obtained in addition to BMI or body weight for diagnostic purposes. Normal weight with abdominal obesity and normal weight obesity must especially be considered in risk assessment. However, the latter also eludes the combined measurement of BMI and waist circumference and can be indirectly determined by metabolic factors [51,52]. 

In order to detect an increased body fat mass or a risky fat distribution, it is neither recommended nor necessary to use elaborate, time-consuming and sometimes radiation-based procedures for routine assessment. Moreover, BMI should not be disregarded because of its limitations, as it essentially provides an initial indication that must then be supplemented by further measurement of fat distribution, e.g., WC or WHtR. In particular, in individuals who do not show an increased risk either by BMI or by their fat distribution pattern, other risk factors, such as those of metabolic syndrome, should be investigated in cases of suspicion.

It is time to raise awareness of the risks of not only visible obesity but also hidden obesity. To this end, a procedure for identifying those at highest risk in routine care must be established which is as standardized as possible.

## Figures and Tables

**Table 1 ijerph-18-11070-t001:** Overview of the most frequently used measures to determine overweight, obesity and increased fat mass.

Measure and Reference	Dimension	Components	Instruments	Cut-Off/Domains of Definition
Body Mass Index (BMI) [9]	kg/m^2^	body weight, body height	stadiometer, calibrated flat scale	≥25 overweight ≥30 obesity
Waist Circumference (WC) [13]	cm	waist circumference	metal measuring tape	≥80 cm f *, ≥94 cm m * ≥88 cm f **, ≥102 cm m **
Waist-to-Hip Ratio (WHR) [13]	cm/cm	waist circumference, hip circumference	metal measuring tape	≥0.85 f, ≥0.90 m
Waist-to-Height Ratio (WHtR) [14]	cm/cm	waist circumference, body height	metal measuring tape, stadiometer	≥0.5
Visceral Fat (VAT, VFA) [15,16]	cm^2^	intra-abdominal fat	DXA, CT	no validated cut-offs
Visceral Fat Thickness (VFT) [17]	cm	intra-abdominal fat	ultrasound	no validated cut-offs
Sagittal Abdominal Diameter (SAD) [18]	cm	intra-abdominal fat	abdominal caliper	19.3 cm f, 20.5 cm m no validated cut-offs
Percentage Body Fat (%BF) [19]	kg/kg	total body fat, body mass	BIA, DXA	37.1% f, 25.8% m ^1^ no validated cut-offs
Body Adiposity Index (BAI) [20]	((HC/height)^1.5^)-18	hip circumference, body height	metal measuring tape, stadiometer	no validated cut-offs ^2^
Body Shape Index (ABSI) [21]	WC/ BMI^2/3^height^1/2^	waist circumference, body weight, body height	metal measuring tape, stadiometer, calibrated flat scale	no validated cut-offs ^3^

NOTE. BMI, body mass index; WC, waist circumference; WHR, waist-to-hip ratio; WHtR, waist-to-height ratio; VAT, visceral adipose tissue; VFA, visceral fat area; VFT, visceral fat thickness; SAD, sagittal abdominal diameter; %BF, percent body fat; BAI, body adiposity index; HC, hip circumference; ABSI, a body shape index; * increased; ** highly increased; ^1^ Cut-off values for 45–64 year olds; ^2^ BAI serves as an estimate of %BF; ^3^ z-score quintiles for relative mortality risk available; f, female; m, male; BIA, bioelectrical impedance analysis; CT, computed tomography; DXA, dual-energy X-ray absorptiometry.

**Table 2 ijerph-18-11070-t002:** Associations of different indices of increased body fat with mortality.

Measure, Reference	Sample	Type of Study	Study Objective	Conclusion
Statements on General Obesity (BMI, %BF)				
BMI [29]	273,843 US-Americans, average age of 38.2 years	retrospective cohort study 1965–2012	risk of death of those over 30 years of age in relation to the BMI baseline value	compared to people with a normal BMI, those with overweight and obesity had an increased risk of death (*p* < 0.001)
BMI [30]	5540 US-Americans (non-smokers), aged 50–84 years	population-based survey 1988–1994, follow up 1999–2004	obesity-related mortality using maximum BMI	using the maximum BMI showed that estimates based on the BMI at the time of the survey can significantly underestimate the mortality burden associated with obesity
%BF, BMI WHR [31]	15,062 Britons from Norfolk, aged 40–79 years	prospective population-based study 1997–2011	using %BF to predict all-cause mortality	when BMI and WHR are considered, %BF does not contribute in prediction
Statements on Indicators of Abdominal obesity (WC, WHtR, WHR, ABSI, VFA, SKA, VSR)				
WC in BMI categories [32]	8,796,759 South Koreans, aged 30–90 years	population-based survey 2009, follow up Ø 5.3 years	relationship between waist circumference and all-cause mortality	abdominal obesity showed a significant but variable relationship with mortality by age, sex, and BMI category
ABSI, HC, WC, WHR, WHtR in BMI categories [33]	352,985 Europeans from 10 countries, aged 35–70 years	prospective cohort study, mean follow up 16.1 years	comparison of alternative abdominal indices to complement BMI in the assessment of all-cause mortality	the highest quartile of the ABSI identified 18%–39% of people within each BMI category who had a 22%–55% higher risk of death
BMI, WC, WHR [34]	15,125 adults with CAD, aged 65.7 ± 11.5 years	5 prospective cohort studies 1980–2008, median follow up 2.3 years	relationship between abdominal (WC, WHR) and general obesity (BMI) and mortality in coronary heart disease	abdominal obesity was also associated with higher mortality in the subset of patients with normal BMI (*p* < 0.001); BMI was inversely associated with mortality
BMI, WC [35]	41,439 Australians from the Melbourne area, aged 27–76 years	prospective cohort study 1990–1994, follow up until 2012	determination of mortality risk and quantification of deaths that are attributable to combinations of BMI and WC	the estimated proportion of all-cause mortality and CVD mortality attributable to obesity as defined by WC alone or BMI and WC was higher than that of obesity as defined by BMI alone
BMI, WC, WHtR, WHR, ABSI [36]	6366 Dutch from Rotterdam, aged > 55 years	prospective population-based study 1989–2002	evaluation of the predictive performance of BMI, WC, WHtR, WHR and ABSI in relation to all-cause, CV and cancer mortality	in the multivariable model, ABSI showed a stronger association with mortality compared to BMI, WC, WHtR and WHR, but the additional predictive benefit was limited
BMI, VFA, SFA, VSR [37]	32,593 South Koreans, mean age 51.3 ± 9.6 years	retrospective cohort study 2007–2015	predictive value of body fat for all-cause mortality	VFA/SFA ratio (VSR) was an independent predictor of all-cause mortality (stronger than BMI, *p* = 0.005)
BMI, BAI, WC, WHtR, WHR [38]	13,307 Germans, aged 25–74 years	prospective population-based study 1989–2002	relevance of anthropometric measurements to cause-specific mortality risk	abdominal obesity was an indicator of higher all-cause and CVD mortality risk
BMI, WC, WHtR, WHR [39]	10,652 Germans, aged ≥ 18 years	1 primary care and 1 population-based cohort study, follow up 3.3–8.5 years	comparison of the association of various measures of obesity with cardiovascular events and mortality	WHtR was the best predictor of cardiovascular risk and mortality, followed by WC and WHR. The use of the BMI is not recommended
BMI, WC, WHR, WSR (=WHtR), ABSI [40]	46,651 Europeans, aged 24–99 years	prospective population-based study in 4 European countries, median follow up 2.5–21.8 years	relationship between CVD mortality and various obesity indicators	indicators of abdominal obesity, such as WC, WHR, WhtR, were stronger predictors of CVD mortality than the general obesity indicator BMI
Statements without a Focus on General or Abdominal Obesity				
BMI, %BF, WC, WHR, WHtR [41]	11,940 US-Americans, aged > 25 years	population-based survey 1988–1994, follow up until 12/2000	comparison of excess mortality associated with different anthropometric variables	attributable fractions of deaths were similar for all measures
BMI, WC, WHR [42]	9603 US-Americans, aged > 18 years	population-based survey 1988–1994, follow up until 12/2000	age-related differences in obesity risk for all-cause mortality	effects of obesity on mortality risk only in adults <65

NOTE. BMI, body mass index; %BF, percent body fat; WHR, waist-to-hip ratio; WC, waist circumference; WHtR, waist-to-height ratio; HC, hip circumference; ABSI, a body shape index; CAD, coronary artery disease; VAT, visceral adipose tissue; VFA, visceral fat area; SFA, subcutaneous fat area; VFR, visceral-to-subcutaneous fat area ratio; BAI, body adiposity index.

**Table 3 ijerph-18-11070-t003:** Associations of different indices of increased body fat with risk of disease.

Measure, Reference	Sample	Type of Study	Study Objective	Conclusion
BMI, WC, ABSI [46]	26,607 Canadians from Alberta, aged 35–69 years	prospective cohort study in 2000, follow up until 06/2017	associations between measurements of body mass and shape and the risk of developing cancer	abdominal obesity appears to be a stronger predictor of overall cancer risk than body mass
BMI, WC, WHR, weight gain [47]	median number of subjects per meta-analysis—1,772,034	umbrella review 204 meta-analyses of cohort studies (73%) and case-control studies	evaluation of the strength and validity of the evidence for the association between obesity and the risk of developing or dying from cancer	association for 11 cancers was supported by strong evidence; the increase in cancer risk per 5 kg/m^2^ increase in BMI ranged from 9% to 56%
WC, WHR, WhtR, ABSI, %BF, BMI [48]	27,557 Swedes from Malmö aged 41–73 years	prospective cohort study 1991, median follow up 19.8 years	which body composition measures have the highest association with the development of hematologic malignancies	measures of abdominal obesity may better predict risk of developing hematologic malignancies, particularly multiple myeloma, compared with BMI
BMI, WC, WHR, WHtR, SAD [49]	6626 Finns, aged 54 ± 15 years	population-based survey 2000–2014	the importance of sagittal abdominal diameter (SAD) as a predictor of liver disease.	SAD provided no additional benefit over WC, WHR and WHtR in predicting cases of severe liver disease; BMI was non-significant
WC, WHR, WHtR, BMI [50]	54,717 Europeans and Australians, median age 52 (male) and 48 (female) years	2 prospective, multi-centre cohort studies 1983–2002	test of the hypothesis that indicators of visceral adiposity (WC, WHR, WHtR) are better predictors of stroke risk than BMI	indicators of abdominal obesity, particularly WHtR, are more strongly associated with the risk of stroke than BMI
BMI, WC, WHR [51]	221,934 persons from 17 countries	58 prospective studies with at least 1 year follow up	to examine the separate and combined associations of BMI, WC and WHR with risk of first-ever cardiovascular disease	all indices showed a similar increased risk, but no significantly improved risk prediction when information on blood pressure, diabetes and lipids was available

NOTE. BMI, body mass index; WC, waist circumference; ABSI, a body shape index; WHR, waist-to-hip ratio; WHtR, waist-to-height ratio; SAD, sagittal abdominal diameter.

## Data Availability

Not applicable.

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
