# Peer review of "The Weight Problem: Overview of the Most Common Concepts for Body Mass and Fat Distribution and Critical Consideration of Their Usefulness for Risk Assessment and Practice"

_ijerph, 2021, doi:10.3390/ijerph182111070_

Round 1

Reviewer 1 Report

Assessing the risk of developing metabolic chronic diseases is essential to maintaining good health in the population. It seems advisable to pay attention to the indicators of excessive accumulation of fat tissue in the body as important elements in education aimed at people at risk of deteriorating health.

For a better comparison and discussion of the issues presented in the review, it would be necessary to:

  • in the methodology:
  • indicate what type of research was used for the analysis? what were the criteria for including studies in the analysis? what was the characteristics of the groups of respondents (age, sex, health condition, etc.)?;
  • a reference to Table 1 should be given;
  • why the criterion of abdominal obesity was adopted - ≥ 88 cm for women and ≥ 102 cm for men, and not more often used ≥ 82 cm for women and ≥ 94 cm for men;
  • Table 1 uses different body fat contents cut points than in the studies description (line 98-99);
  • it would be worth describing and discussing the specificity and specificity of the methods of body composition determination, it is decisive for the possibility of their use in practice;
  • in order to conclude on the usefulness of obesity indicators, shouldn't we also discuss the ways of defining mortality and risk of disease in the studies?;
  • presenting the characteristics of people participating in the described studies is particularly important in the discussion of the possibility of using the following definitions: Normal Weight Obesity, Normal Weight Abdominal Obesity and Metabolically Healthy Obesity in the risk analysis of deteriorating health condition;
  • in the discussion of the results of the presented studies, the definition of the indicator: maximum BMI in the observed period and the possibility of its use in research should be clarified;
  • it is worth discussing the impact of age and disease-related changes in weight on the use of BMI as a risk factor for increased mortality;
  • the conclusions refer to the possibility of using bioimpedance as a method of determining body composition, but the article lacks an analysis of the usefulness of this method compared to other methods of determining body fat content;
  • Table 4: How was Disease Risk determined - What methods were used to determine them and what diseases were the risks identified?;
  • the conclusions indicate the limitations of waist circumference measurement (line 242-244) - what are the limitations of other methods of anthropometric measurements? Did the above-mentioned studies point to these limitations?

Reviewer 2 Report

  • Given the contents of the paper, I would ask that this be placed as a review rather than a viewpoint.
  • Line 30,47,50,91,120: Please define the types of study design.
  • Line 40-43: It is not clear why authors focused on Germany here. How about other contexts?
  • The introduction is short. I would like the authors' views on these questions: Is there a Gold Standard in body fat measurement? What are the types of ways body fat percentages can be measured? What are the problems/challenges with the accuracy of measuring BMI in children and adults? What are the techniques used to measure the trends/levels in obesity-attributable mortality?
  • Line 61-66: Which period does the literature search cover? The types of study design should also be clearly mentioned (i.e., cross-sectional, longitudinal…..etc).
  • Table 1: needs to be clarified. It is not organized. For example, the first column can be omitted, and replace "acronym" with "measure and References".
  • Line 82-101: I think these paragraphs should not be placed here. Authors should discuss the advantages and disadvantages of using these instruments? Which instrument gives more precise calculation of different measures?
  • Table 2: Should be study "objective" not "objektive".
  • Table 2 & 3: Significant P-values should be included in the conclusion column.
  • Table 2: What does "general statements" mean?
  • Line 112-126, 135-137: These paragraphs could be improved. I miss a discussion on the main results in Table 2.
  • Section 8 (Line 195): It would also be helpful if a sentence or so could be added after a paragraph that what is the message from the information compiled in this section.
  • The conclusion is very weak in its current form and need to be strengthened and expanded.
  • Line 240: I think Table 4 should not be placed here.

Reviewer 3 Report

The viewpoint by Kesztyüs and colleagues addresses the topic of body mass index (BMI) and obesity related outcomes. Overall, the review is well written. The major concern is the interest to readers, the viewpoint expressed by the authors is very much in line with conventional wisdom in the field. This will have very little appeal to the practicing clinician or obesity research.BMI will continue to be used an index for adiposity and most clinician incorporate waist circumference into their patient assessments.

One way to improve the reader interest is to provide the hypothesis or well- articulated viewpoint in the Introduction.  As it stands, the "In the following, the BMI and the most important concepts of alternative determinations of excessive body mass or body fat mass are presented and discussed with regard to their informative value in terms of health risk. Emphasis is placed on the applicability of the measures in clinical practice.". There is nothing provocative about the state purpose and will have low interest to the field.

Round 2

Reviewer 1 Report

The authors of the manuscript provided factual and comprehensive responses to all comments in the review. They compared and discussed the use of different methods and indicators in the assessment of obesity. They also analyzed the possibilities of their use in assessing the risk of cardiovascular disease and cancer. In this version of the manuscript, clearly formulated conclusions that correspond to the subject of the presented analysis deserve attention.

The manuscript presented for review may be published in its current form.

Reviewer 2 Report

No further comments.

Reviewer 3 Report

I have no other concerns